**communications** engineering

# Fast automatic multiscale electron tomography for sensitive materials under environmental conditions
Louis-Marie Lebas [1], Karine Masenelli-Varlot [1], Victor Trillaud [1], Cédric Messaoudi [2], Mimoun Aouine[3], Laurence Burel[3], Valentine Noblesse[1,4], Clémentine Fellah[4], Erwan Allain[1], Christophe Goudin[1], José Ferreira[1], Matthieu Amor[4] & Lucian Roiban [1] ✉

The demand for characterisation of beam-sensitive samples at the nanoscale in environmental conditions is increasing for applications in materials science and biology. Here we communicate a protocol with custom software, enabling precise control over the electron microscope, and a custom sample holder, facilitating automated acquisition of fast 3D data from a single object under environmental conditions. This method enables imaging with a controlled electron dose and multi-modal electron signals. The method can be used in environmental scanning or transmission electron microscopes for easy sample preparation and to benefit from high spatial resolution, respectively. To demonstrate its effectiveness, we investigate the porosity of $Al(OH)_3$ hydrogels, and the penetration ability and distribution of gold nanoparticles. Unfixed, hydrated magnetotactic bacteria producing intracellular iron oxide nanoparticles were also characterized in 3D in their native state. This methodological and technical development serves as a milestone in the study of various samples at any humidity level, offering easier sample preparation compared to cryo-TEM techniques, while maintaining a similar or even lower dose level.

Cryo-electron tomography (Cryo-ET) is now standard for 3D imaging of biological samples at the nanometer scale[1–3]. It leverages tilt-series acquisition, useful for analyzing large, complex single objects[4]. A key advancement is Cryo-ET in scanning transmission electron microscopy (STEM) mode[5–7]. STEM provides high spatial resolution and contrast for biological materials. It achieves a lower electron dose rate compared to conventional TEM, for the same final cumulative dose, by enabling precise control over the dwell time at each scanned point, which minimizes sample damage and optimizes imaging conditions[8]. However, the cryogenic method prevents the study of liquid water samples under dynamic or environmental conditions, and sample preparation remains time-consuming and complex[9]. In recent years, the environmental mode has allowed imaging of electron-sensitive samples in quasi-native or liquid states by creating partial gas pressure inside the microscope chamber[10,11]. Alternative methods, such as liquid cells[12–14], preserve the liquid state in a small volume but offer limited tilt angles[15] and make dynamic control of the environmental composition more challenging. Recent experiments have shown promising results in acquiring 3D tilt series of hydrated samples under environmental conditions. Wet-STEM

tomography has been demonstrated in environmental scanning electron microscopy (ESEM)[16–18]. Environmental transmission electron microscopy (ETEM) has reported 3D environmental tilt series, but not in fully hydrated conditions[19], or not in STEM mode[20]. Despite these advances, challenges persist in managing electron dose, achieving spatial resolution, and maintaining stable water layer thickness. Minimizing electron dose to avoid beam damage while meeting spatial and temporal resolution requirements is crucial. These complexities require a high degree of automation and control over the microscope and sample holder.

We propose here a methodology and software to automate fast 3D data acquisition of a single object under environmental conditions, with high spatial resolution, controlled low electron dose, in STEM, and with multi-mode electron signals. Our method suits various scales for both ESEM and ETEM. We validate our approach using aluminum hydroxide hydrogel $Al(OH)_3$ vaccine adjuvants with gold nanoparticles. Gold acts as a strong electron scatterer in STEM mode[21], while $Al(OH)_3$ is beam-sensitive and widely studied[22]. We specifically examine hydration and dehydration cycles by analyzing gold distribution and nanoscale porosity within $Al(OH)_3$

[1]MATEIS, UMR5510, Univ Lyon, INSA Lyon, UCBL, CNRS, Villeurbanne Cedex, 69621, France. [2]Multimodal Imaging Center, Institut Curie, CNRS UAR2016, INSERM US43, PSL Research University, Université Paris-Saclay, Orsay, France. [3]IRCELYON, UMR 5256, CNRS et Université Lyon 1, Villeurbanne Cedex, 69626, France. [4]ENS Lyon, UMR 5276, LGL-TPE, Université Claude Bernard Lyon 1, CNRS, Lyon, 69626, France. ✉e-mail: lucian.roiban@insa-lyon.fr

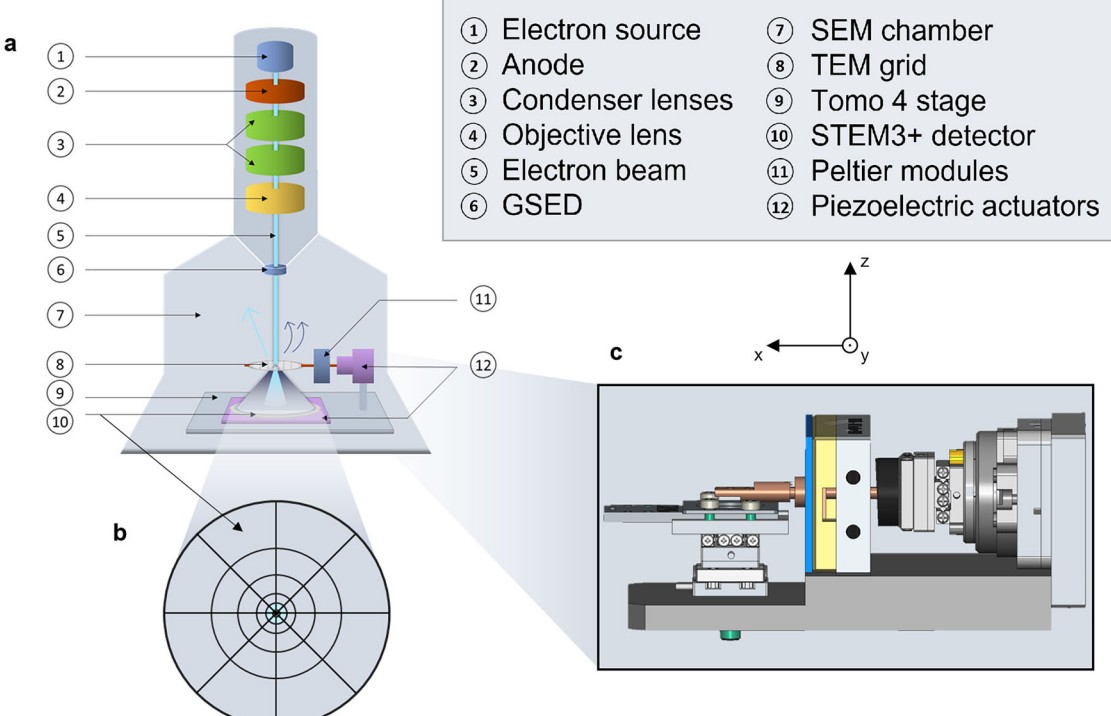

**Fig. 1 | Comprehensive overview of the experimental setup for STEM liquid-phase tomography in ESEM. a** Illustration of the environmental electron microscope (ESEM) column with the sample stage inside. **b** XY representation of the STEM3+ detector with separated segments for versatile acquisition modes: BF or ADF. **c** XZ visualization of the home-made tomographic stage "Tomo 4", comprising piezo-inertial elements and Peltier modules.

hydrogels. Finally, our methodology is used to characterize magnetotactic bacteria *Magnetospirillum magneticum* strain AMB-1, which synthesizes intracellular chains of ferrimagnetic nanoparticles made of magnetite ($Fe_3O_4$) in a genetically controlled manner[23]. In these bacteria, the magnetite crystals are produced in dedicated organelles called magnetosomes, which consist of a bi-layered lipid membrane surrounding a single nanoparticle[23]. The bacteria are analysed in their native state, i.e. in diluted growth medium without dehydration, fixation or stain.

## Results
### Experimental setup and simulation
Experiments are conducted using a ThermoFisher QuattroS ESEM and a ThermoFisher Titan 80−300 kV ETEM. The ETEM employs a commercial Gatan Elsa cryo sample holder, whereas an evolved version of the home-made tomographic stage "Tomo 3" from our lab is used[24]. This prototype has been effective for electron tomography of latex particles in water[25], manually acquiring tilt-series with up to a dozen images.

Figure 1a shows the ESEM setup, highlighting the chamber size to accommodate the stage and key microscope components. The stage design (Fig. 1c) supports tilt-series acquisition in environmental conditions, with precise control over temperature and eucentric position. Sample hydration is managed by adjusting temperature with Peltier modules and water vapor pressure via the microscope, following the dew curve of water (Fig. 2c).

The eucentric position is controlled by two linear piezoelectric actuators, with the stage mounted on a 360° rotating piezo controller. Images are recorded using two detectors: a GSED for surface characterization[26] and a STEM3+ detector (Fig. 1b), which collects transmitted and scattered electrons in bright field (BF) and annular dark field (ADF) modes, respectively. The latest ESEM stage, "Tomo 4" Supplementary Fig. 1, allows moving and centering the STEM3+ detector under the sample with piezo-inertial elements.

Preliminary Monte Carlo simulations assess the electron transparency of the sample. Figure 2a, b depict results for a sample covered by a water layer. The simulation shows the transmitted electrons in BF and ADF modes for $SiO_2$ with various thicknesses and water layers. It reveals material transparency up to 2 μm with a 350 nm thick water layer (arbitrary 5% condition[24]). BF shows a transparency decrease with increased material thickness, while ADF shows more scattering in thicker specimens. This can cause contrast inversion between thin and thick parts, as noted in[24]. Combined with[27] on spatial resolution in liquid STEM, the simulation demonstrates the potential for adequate contrast and spatial resolution for studying hydrated samples below 10 nm.

### Software for automatic Liquid 3D STEM tomography
To meet demands for accuracy, speed, efficiency, and user-friendliness, we developed M-SIS software for automatic tilt-series acquisition. It provides a simple workflow for eucentric setup and acquisition initiation. Implemented in Python for ease of development and maintenance, it integrates native libraries like Thermo Scientific™ Autoscript for ESEM and DigitalMicrograph™ for ETEM to control microscopes. Smaract MCS-3D piezo-inertial elements require a C++ library. Key features include automatic eucentricity, drift correction, automatic acquisition, and focus/astigmatism correction assistance. Figure 3 shows the software workflow.

Achieving precise eucentric positioning and aligning the region of interest (ROI) can be challenging. Addressing the issue raised by ref. 28 on automatic eucentric position adjustments in electron tomography, we present a fast calibration method to correct eucentric positioning using live imaging measurements. More precise calibration reduces post-processing for 3D reconstruction and enables faster 3D STEM-in-ESEM imaging[29]. Our automatic eucentricity method uses the shift in 3D feature positions in 2D images as the specimen tilts, along with a theoretical model (protocol in Methods), to compute correction parameters.

Once the ROI is at the eucentric position, tilt-series acquisition can start. However, drift occurs at high magnification due to factors like eucentric inaccuracy[30], membrane movement[31], thermal movement[32], and charging effects[33]. Stage mechanical defects may also contribute[34]. While the

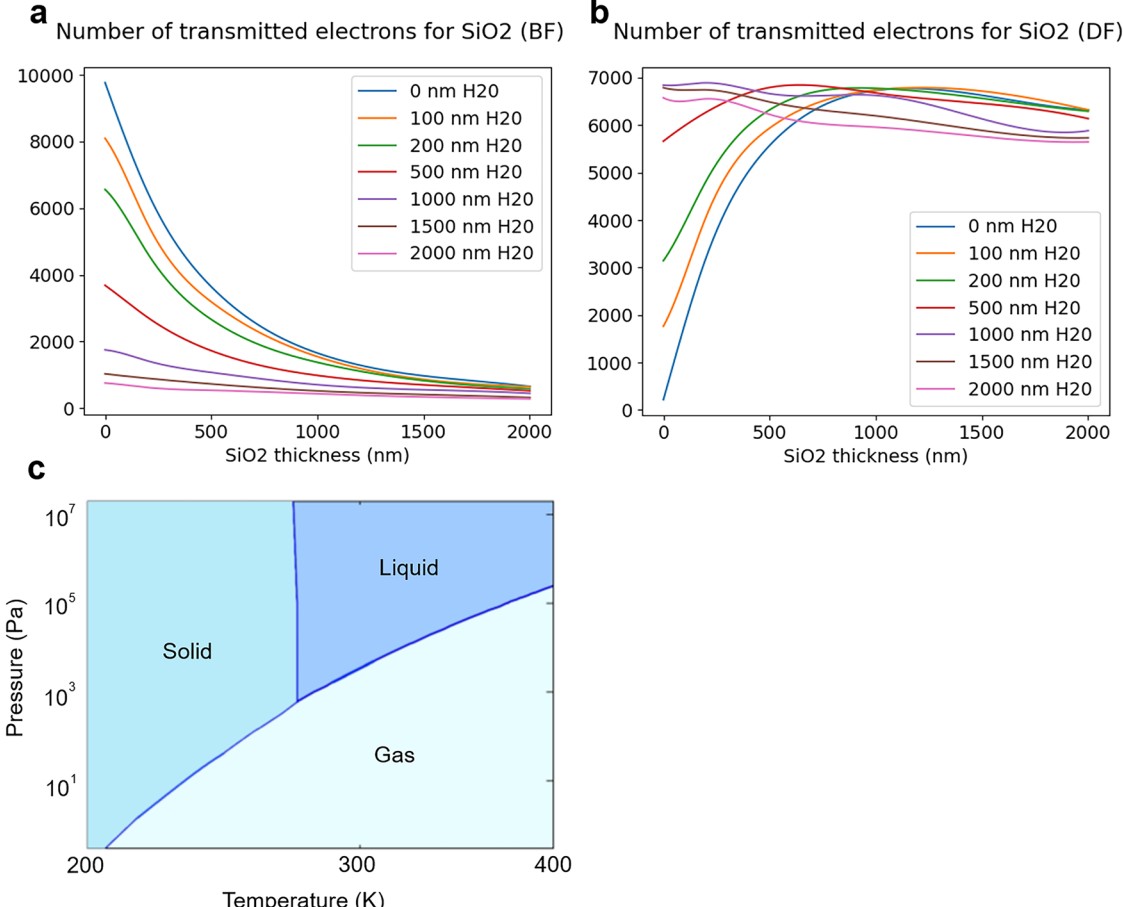

**Fig. 2 | Simulated graphs of transmitted electrons and water phase diagram.**
**a** Simulated number of transmitted electrons (BF mode) for different thicknesses of
$SiO_2$ and different water layer thicknesses. **b** Simulated number of transmitted
electrons (ADF mode) for different thicknesses of $SiO_2$ and different water layer
thicknesses. **c** Gas pressure and sample temperature are used to control the hydra-
tion state of the sample.

liquid layer may cause drift from gravity flow and Brownian motion, this is
often negligible when the layer is thin[25]. Drift correction methods have long
been used in microscopy, primarily relying on image analysis with cross-
correlation[35], feature matching[36], or similar algorithms[37]. After detecting a
shift, correction is applied, followed by a validation step with another image
to confirm accuracy. If needed, further corrections are made. In electron
tomography on beam-sensitive specimens, this validation step increases
acquisition time and electron dose. A common workaround is using a
different region for correction, but extended acquisition time can impede
dynamic sample analysis. Higher stability is also vital for studies with liquid
or living biological samples.

Avoiding the validation step reduces the total dose received by the
sample. However, a simple analysis-correction pair without validation
would still diverge, though more slowly (see simulation below). Our method
follows a different approach, eliminating the need for validation. It not only
corrects detected drift but also anticipates future drift in subsequent
acquisition steps (i.e., next image with time increment and next angle).
Using previous states and corrections, the software estimates a linear drift
model, adjusting continuously and retroactively accounting for drift
changes.

Before applying the workflow, we simulated simple scenarios using
equations from the Methods section to assess the algorithm's performance.
We considered three cases with integral drift and three with random drift
ranging between $[-0.5, 0.5]$ pixels per image. The first case shows drift
correction with linear integral drift, the second examines behavior with
straight line segments, and the third involves quasi-quadratic drift. Random
drift is common in microscopy at short time scales[35,36]. Results in Fig. 4

demonstrate our solution effectively corrects drift, keeping the ROI cen-
tered. Although robustness to random drift is good, corrected positions
sometimes briefly exceed the acceptable range of $[-1, +1]$ pixels. Occa-
sional stabilization cycles introduce slight oscillations. Drift in the
z-direction can also be addressed with more difficulty (see Methods).

Finally, the design of "Tomo 4" allows simultaneous use of multiple
detectors during acquisition. This enables accessing complementary infor-
mation from the same electron paths. Using M-SIS software, tilt series can be
acquired simultaneously in SE with a GSED and in BF and ADF with the
modular STEM3+ detector.

In ETEM, the sample is placed on a cryo holder with temperature
controlled remotely using liquid nitrogen and a resistor. Unlike in ESEM,
specimen holders cannot be introduced under humid conditions due to
non-customizable pumping in the microscope entrance chamber. Thus, the
droplet must be placed on the grid on the holder and the protective tab
closed to prevent drying. To ensure gradual cooling and avoid abrupt
temperature drops that could affect stabilization and freezing, the liquid
nitrogen tank can be pre-cooled with a few centiliters in advance to mini-
mize thermal fluctuations. Once stable conditions are achieved, droplet
thinning and alignment are done, and tilt-series acquisition with M-SIS is
performed as in ESEM.

**Multiscale analysis of aluminum hydroxide gel**
The liquid-phase STEM tomography method is validated using commercial
aluminum hydroxide gel, a common vaccine adjuvant[38]. Its 3D structure,
high porosity, and electron beam sensitivity make it ideal for testing. By
depositing gold nanoparticles (10 nm in diameter) onto the gel, we evaluate

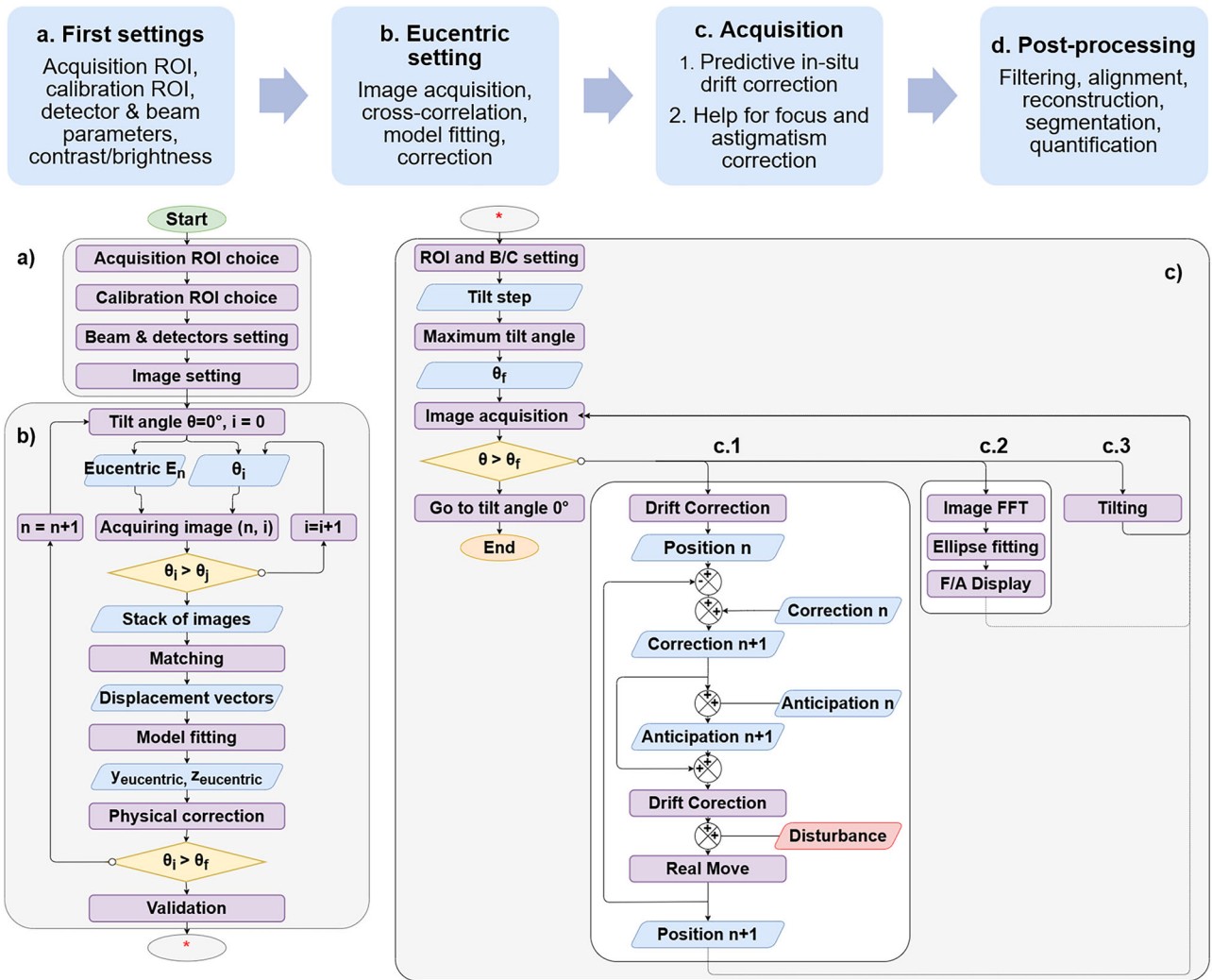

**Fig. 3 | Overview of the liquid-phase 3D STEM tomography workflow for automatic tilt-series acquisition. a** Establishment of initial parameters: region of interest (ROI), beam configuration, and image setup. **b** Eucentric setting. **c** Sequential acquisition of the tilt series. **c.1** Predictive in situ drift correction. **c.2** Assistance with focus and astigmatism correction. **c.3** - Tilting of the sample holder. Tasks (**c.1**−**c.3**) are performed in parallel.

their penetration and distribution. Gold beads can also mark target antigens[39]. Combining the capabilities of both environmental microscopes, we characterize the gold nanoparticles' spatial distribution and the porosity of $Al(OH)_3$ hydrogel. The size of the gold particles and their ease of observation enable characterization by ESEM, while ETEM provides high-resolution quantification of porosity.

In ESEM, the temperature is set to 1°C. Gold nanoparticles were deposited onto a 400-mesh TEM copper grid with a holey carbon membrane to aid tilt-series alignment and improve tomogram reconstruction quality. The sample was prepared by diluting the solution in deionized water and depositing a droplet (<10 μL) on the grid. Purge cycles between 12 and 16 hPa were used to prevent dehydration. To avoid radiation damage, a preliminary test estimated the critical dose for degradation, with visual indicators like morphological changes and shade variations[40]. We used an acceleration voltage of 30 kV with a spot size of 3 for electron transparency. The dwell time ranged from 1−5 μs, with a resolution of 1536 × 1024 to enhance the signal-to-noise ratio and minimize acquisition time. The horizontal field of view spanned one to a few micrometers. After purging, the pressure was decreased to evaporate water and make the sample electron transparent[18]. During this, water evaporation cools the sample, while condensation warms it[41]. Temperature monitoring identified the regime: evaporation, condensation, or stability. After preliminary

adjustments, including beam, focus, and astigmatism alignments, hydrated particles with visible gold nanoparticles were selected as ROIs.

The eucentric was set using the M-SIS software, in an area far away from the ROI to avoid excessive electron dose, on the tilt axis but with a different x-position. After adjustments, a tilt series was acquired on the ROI with a dwell time of 5 μs, a pixel size of 5.5 nm, from − 70° to + 70° (tilt step 1°), plus additional images at higher angles. Following a 5-min drying step with reduced pressure, a similar tilt series was acquired on the same ROI with the same settings. Lastly, environmental conditions for the hydrated state were restored in 5 min, and a final tilt series was acquired.

Figure 5a shows the tomograms obtained. The spatial resolution achieved, assessed via the Fourier Shell Correlation (FSC) curve[42,43], ranges from 26 to 50 nm based on the 0.143 and 0.5 criteria. This resolution decreases along the z-axis due to noise. The electron dose per image is $9\,e^-\cdot nm^{-2}$ [44]. For the entire tilting series in ADF mode, the total dose per tomogram is between 1050 and $1250\,e^-\cdot nm^{-2}$. Acquisition of the BF signal beyond the cut-off angle where the electron beam is blocked by the TEM grid bars is not feasible, unlike in ADF mode, enabled by electron scattering. Thus, the choice is between BF-only acquisition (fewer images, lower dose) or combined BF and ADF acquisition (more images, complementary reconstructions). Regardless of the mode, the total dose remains below $3500\,e^-\cdot nm^{-2}$.

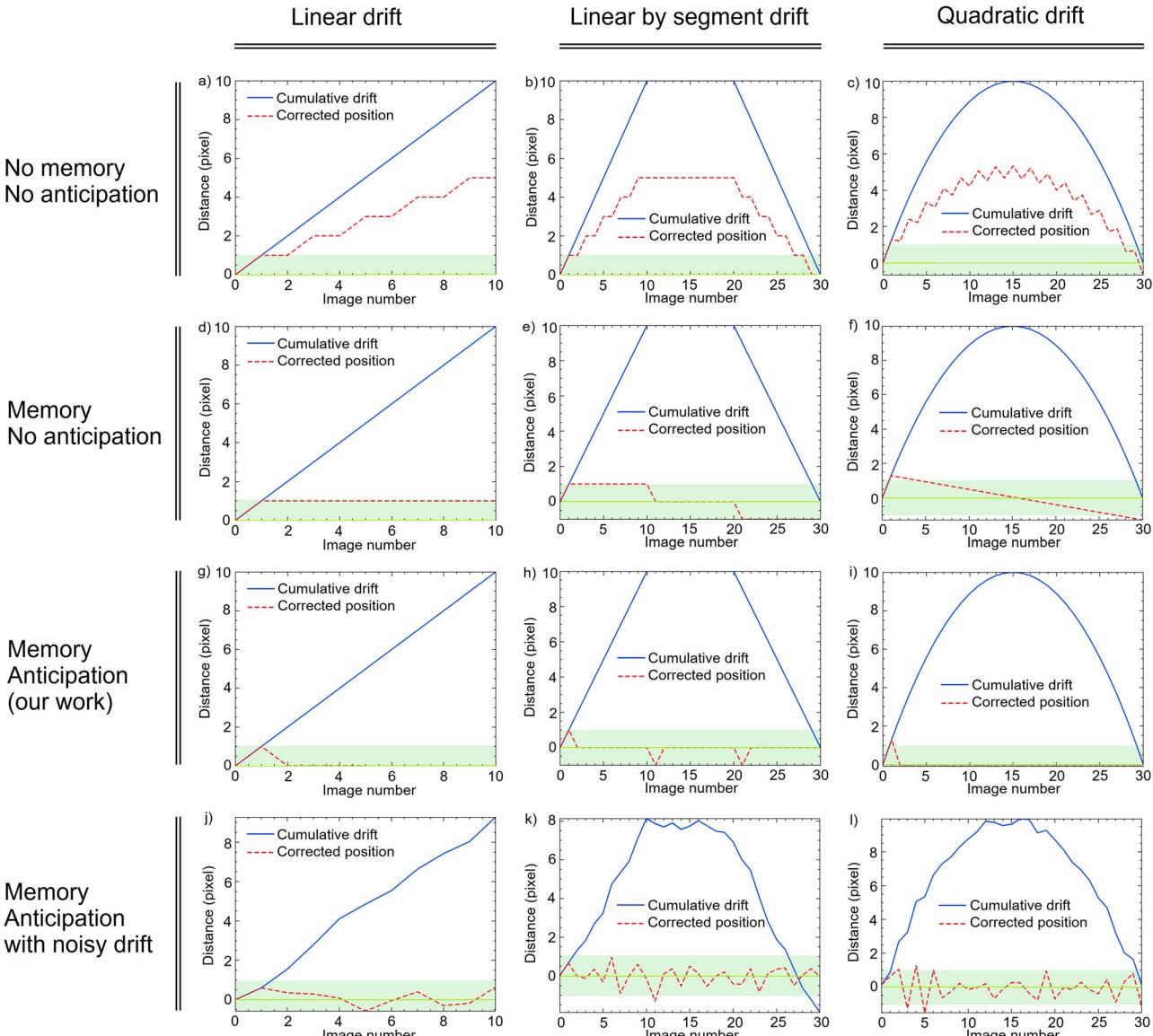

**Fig. 4 | Simulation of the predictive in situ drift correction algorithm.** The blue curve represents the cumulative drift, while the dashed red curve represents the corrected position. The first row shows the drift correction with correction only: (**a**) Linear noise-free drift. **b** Linear segment-wise noise-free drift. **c** Square-law noise-free drift. In the second row, the drift correction incorporates both correction and memory of previous moves, without anticipation: (**d**) Linear noise-free drift. **e** Linear segment-wise noise-free drift. **f** Square-law noise-free drift. The third row illustrates the drift correction with correction, memory, and anticipation: (**g**) Linear noise-free drift. **h** Linear segment-wise noise-free drift. **i** Quadratic-like noise-free drift. Finally, the fourth row depicts the drift correction with correction, memory, and anticipation, including random noise ranging from [− 0.5, 0.5] pixels for each subsequent image: (**j**) Linear drift with noise (**k**) Linear segment-wise drift with noise. **l** Quadratic-like drift with noise. The green strip indicates the acceptable zone where the drift is not visible in the image due to its magnitude being less than one pixel.

The dehydration-rehydration cycle does not affect the particle's overall structure. Dehydration causes a $3.4 \pm 1.3\%$ shrinkage, followed by a $1.2 \pm 1.0\%$ expansion upon rehydration. Gold particles on the membrane are easily distinguished from those inside the sample. Despite the initial uniform distribution, a statistical analysis of the pair correlation function $g(r)$ (Fig. 5) shows slight clustering, significant when $g(r)$ exceeds the upper boundary of the statistical envelope for random correlation. Although the differences between the 3 curves are minor, clustering is slightly more pronounced in the dry volume, aligning with the observed slight volume decrease during drying.

As the spatial resolution in ESEM is insufficient to quantify sample porosity at the nanometer scale, we applied the same methodology in ETEM. With the higher acceleration voltage in ETEM (300 kV), the theoretical maximum resolution improves, though with a higher electron dose to the sample. After inserting the sample holder, the $H_2O$ vapor pressure is set

to $8.0 \pm 0.1$ mbar and the temperature to $1 \pm 0.5°C$ to maintain a stable hydrated state.

Figure 5b shows the tomogram from a tilt series of 260 projections acquired between $-65°$ and $65°$ with a $0.5°$ tilt step in 12 min. Each $1024 \times 1024$ image had a 0.6 nm pixel size, 1 μs dwell time, and a spot size of 9. Resolution, estimated using FSC, ranges from 6.6 nm (0.143 criteria) to 8.9 nm (0.5 criteria). According to ref. 27, this resolution is close to the theoretical maximum for a liquid water-based layer. The electron dose, based on ref. 45, is $61 \pm 3\,e^- \cdot \text{nm}^{-2}$ per image, totaling $1.6 \pm 0.1 \times 10^4\,e^- \cdot \text{nm}^{-2}$ for the tilt series.

The enhanced resolution compared to ESEM now enables accurate quantification of pore size distribution, see Fig. 5b. The curve reveals a uniform distribution of pores from 5 to 30 nm, reflecting the size of the aluminum hydrogel substructure[46,47]. Beyond 30 nm, the curve indicates varied particle sizes and morphology, up

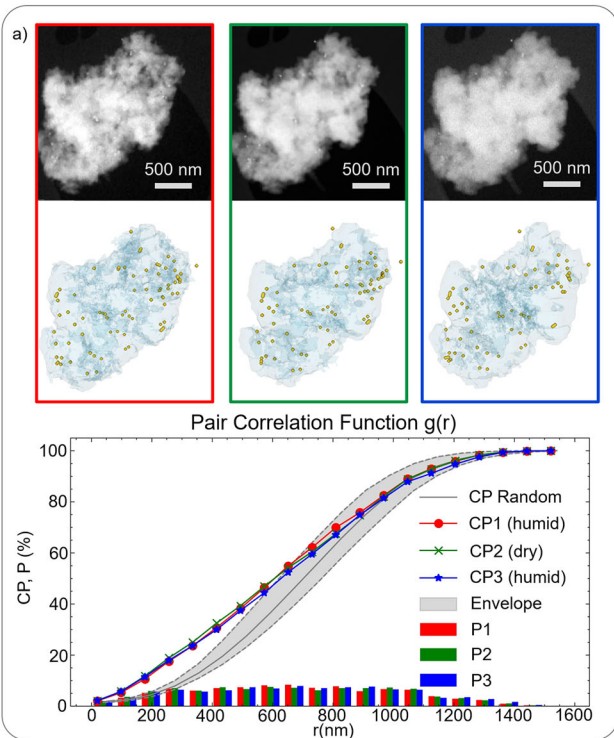

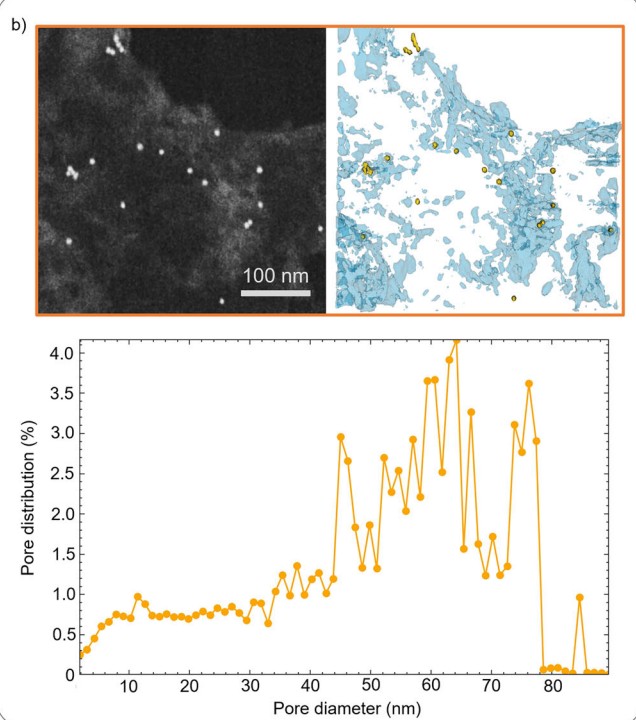

**Fig. 5 | Overview of 3D results and analysis. a** ESEM characterization. Each panel represents an image from the tilt series taken around 0°, along with an XY view of the resulting 3D model. The color scheme represents different states: initial humid state (red), dry state (green), and second humid state (blue). The gold beads are displayed as yellow spheres. Additionally, the Pair Correlation Function g(r) is presented, where r represents the distance between two beads. The figure shows that g(r) values exceed the upper boundary of the statistical envelope corresponding to random correlation, hence revealing a subtle clustering effect. *P* stands for Percentage, and CP for Cumulative Percentage. **b** ETEM characterization. The panel shows an image from the tilt series around 0°, accompanied by an XY view of the 3D model. The pore size distribution obtained from the model shows a uniform distribution of pores between 5 and 30 nm. The values above 30 nm reflect the dispersion on the TEM grid and not the pore size within the hydrogel particles.

to the sample's thickness, which does not provide useful information on the hydrogel's porosity.

### Analysis of hydrated bacteria

AMB-1 bacteria were cultivated under standard conditions, and deposited onto a 400-mesh TEM copper grid coated with a holey carbon membrane. 2D Liquid phase analysis of AMB-1 in ETEM yielded biological samples with preserved typical spirillum shape, size and ultrastructure Supplementary Fig. 6. Similarly to conventional STEM observations (Supplementary Fig. 6a) of dried bacteria using SEM, magnetite nanoparticles could be observed and their chain structure is conserved. The chain organization of magnetosomes is enabled by a complex interplay of proteins that prevent magnetite from collapsing into agglomerates[48,49]. Maintaining the chain structure thus illustrates the preservation of the proteic and lipidic components of magnetosome chains. In addition to these magnetic organelles, AMB-1 also produces protein-bounded intracellular inclusions of organic carbon made of polyesters (polyhydroxyalkanoates or PHA) that can be observed using conventional TEM and STEM as sphere-like structures possessing brighter contrast than the rest of the cell[50]. Those structures are visible in liquid STEM (Supplementary Fig. 6b, c).

The procedure was applied for the analysis of the fully hydrated AMB-1 bacteria in 3D in ESEM. 20 nm diameter Au nanoparticles serving as fiducial markers were deposited on 200-mesh copper grids covered with a holey carbon film. Then, a droplet of 3 μl of diluted fresh growth medium containing the bacterium was dropped on the other side of the grid, and inserted in the ESEM. It must be mentioned that the bacteria were alive and they underwent no treatment - dehydration, fixation, staining—before insertion in the electron microscope. The microscope was used at a high voltage of 30 kV, the spot size was set to 3 and the dwell time to 2 μs. The M-SIS software was used to record a tilt series between ± 60° with a step of

0. 5°. The stack had 242 images with a resolution of 1536 * 1024 pixels, a pixel representing 5.18 nm. After the experiment, the survival of the bacteria was not verified as it falls outside of the scope of this article but, as mentioned above, the ultrastructure of the bacteria was preserved.

The tilt series was aligned by combining TomoJ and Imod, and the volume was computed in TomoJ using 15 iterations of O-SART algorithm with FISTA optimisation[51–54]. For the image alignment only 213 images were preserved. The images that were blurred or in which the bacterium was not fully in the field of view were deleted. Cross-sections of the reconstructed volume are shown in Fig. 6. Liquid, in light gray, is present around the bacterium. Within the bacterium, the magnetite nanoparticles can be distinguished, as well as the PHA inclusions and the wall that separates them. The electron dose was $9.79 \ e^- \cdot nm^{-2}$ per image, therefore a total electron dose of $2.36 \times 10^3 \ e^- \cdot nm^{-2}$ was received by the sample.

### Discussion

We have demonstrated the feasibility of acquiring in situ tilt-series images of beam-sensitive materials in liquid-phase STEM within minutes and with a low electron dose, depending on the acquisition parameters. We developed the M-SIS Python software for tilt series acquisition, which performs automatic eucentric and predictive drift correction while providing real-time information on acquisition quality.

For the first time, we investigated the nanoscale porosity of $Al(OH)_3$ in its hydrated state using an electron microscope. We achieved 3D models with sufficient resolution to examine sample morphology and porosity at the nanoscale. Previous studies were limited to microscale[55] or used bulk dry particles with SEM[46]. The distribution aligns with results from other techniques, such as nitrogen adsorption-desorption BET isotherms[46,47]. Magnetotactic bacteria were also successfully investigated. Using our methodology, we have been able to reconstruct the volume of hydrated

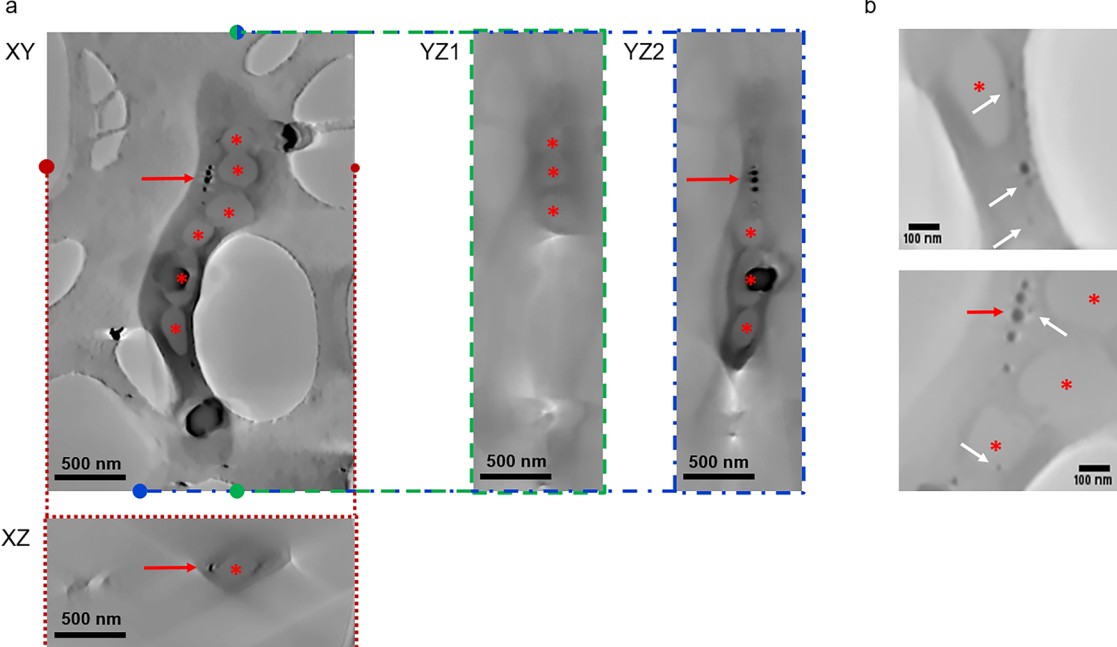

**Fig. 6 | Overview of 3D results for a hydrated bacterium. a** Cross-sections of the reconstructed volume of a magnetotactic bacterium *Magnetospirillum magneticum* strain AMB-1. The volume was denoised using standard non-local means denoising algorithm. Red arrows and stars point to the magnetite chains and PHA granules in the bacteria, respectively. These intracellular structures are preserved, due to the liquid environment. **b** Cropped cross sections from the reconstructed volume. The white arrows indicate 15 nm intrabacterial substructures, that do not appear to be artefacts of the reconstruction process. They likely correspond to growing nanoparticles that are commonly observed in laboratory cultures of magnetotactic bacteria[71].

AMB-1 cells. Previous works have reported cryotomography of AMB-1 coupled to TEM[48,49]. In these cases, the magnification achieved was sufficient to resolve the cellular membranes, including the magnetosome vesicle. Yet, they required complex sample preparation and freezing of the bacteria in liquid ethane[49]. The benefit of our methodology is to enable observation and reconstruct the ultrastructure of native hydrated bacteria. Compared with cryo-electron tomography, not all intracellular features are presently visible in the volume. Nevertheless, the PHA inclusions are clearly visible without any preparation artifact, and features as growing nanoparticles, of size 15 nm, are visible.

Furthermore, we achieved precise control and quantification of the electron dose. The M-SIS software facilitates the acquisition of a nearly continuous tilt series, with enhanced speed and accuracy compared to manual acquisition, enabling the acquisition of a larger number of projections with smaller tilt steps. In comparison, with the manual ESEM procedure, Xiao et al. acquired only 10 – 15 projections for each acquisition in a much longer time, without dose control, resulting in artefacts in the reconstruction[29].

Radiation-induced damage is an important constraint, limiting the deposited electron dose before irreversible damage occurs[56–58]. Water generates free radicals and chemically unstable species that may react with the sample[59]. The literature estimates the critical dose to be between $10^3$[60] and $5 * 10^3$ e⁻· nm⁻²[61,62] for closed liquid cells, to prevent nucleation in liquid samples[8] and avoid exceeding the damage threshold for non-fixed biological samples. Fixed biological samples could withstand up to $10^5$ e⁻· nm⁻²[60,63,64]. The dose rate in the experiments reported here was lower than in previous studies. Specifically, it was about 1.1 e⁻· nm⁻² · s⁻¹ in ESEM and 40 e⁻· nm⁻² · s⁻¹ in ETEM, compared to 4.4 e⁻· nm⁻² · s⁻¹ in ESEM[18] and $3.8 * 10^3$ to $4.8 * 10^3$ e⁻· nm⁻² · s⁻¹ in conventional ETEM tomography[65]. Despite many projections in the tilt series, the total electron dose remains low, given the large number of projections. Although these values exceed the proposed critical dose for biological samples, they are compatible with the sensitivity of the samples shown here. Further reductions in dose can be achieved by decreasing the number of projections, potentially keeping it

below the $10^3$ e⁻· nm⁻² threshold in ESEM. Nevertheless, it should be pointed out that this value may not be a strict limit, especially in our set-up where maintaining thermodynamic liquid-vapor equilibrium allows a continuous supply of fresh water molecules.

The primary limitations of our approach are the thermal and mechanical stability of the experimental setup. Achieving thermal stability is essential to maintain the liquid state and control the liquid film thickness precisely. Both ESEM and ETEM struggle with temperature control due to design complexities. Mechanical stability issues arise from vibrations in the water circulation system of ESEM, which can affect the specimen holder despite damping efforts. Both ESEM and ETEM would benefit from a more precise mechanism for nanometer-scale eucentric point adjustment to reduce drift tracking and enhance acquisition speed. Currently, these limits restrict the liquid layer's stability to a few tens of minutes or hours and prevent reaching resolutions far below 10 nm. Acquisition speed is constrained by detector and computer readout speeds, affecting maximum frequency. Additionally, balancing acquisition speed, number of projections, signal-to-noise ratio (SNR), and reconstruction performance is crucial. Using denoising algorithms may improve SNR while maintaining low dose and high acquisition speed[27]. Although our method simplifies sample preparation with just one droplet on the TEM grid, further investigation into droplet volume variation during thinning is needed.

Until now, liquid phase STEM tomography was limited by its complexity, requiring specialized hardware, precise experimental control, and careful management to avoid sample over-irradiation. We introduce tools and methods that integrate these elements, offering a versatile, comprehensive, and accessible analysis technique. This advancement facilitates the study of unfixed hydrated or immersed biological materials in their native state, complementing existing techniques in material science and biology. This complementarity is obvious in terms of stability, high-angle tomography, hydration and environmental conditions, in both dynamical and static analysis. Adding liquid STEM tomography to techniques like closed cell or cryo-STEM creates a coherent suite for characterizing hydrated

biological samples. Adjusting acquisition parameters can keep the deposited dose below the viability threshold for biological cells[60], making it promising for studying cellular behavior. In this area, Bekel et al. gave a proof of concept with the manual acquisition of a series of tilted images of a hydrated, fixed—not not stained—mouse fibroblast cell, in a buffer medium[18]. Automation will allow faster acquisition with more projections, reducing artefacts and thus improving the quality of the 3D model. The next key step is overcoming temperature constraints to achieve room temperature 3D imaging of biological samples at the nanoscale.

## Methods

### Automatic alignment

The displacement ($a$) of the point of interest $A$ in the acquired images, as depicted in the Fig. 7 can be determined using trigonometric considerations:

$$a = y_A * (1 - cos(\theta) + z_A * sin(\theta)) \tag{1}$$

where $y_A$ and $z_A$ are the coordinates of the point of interest A at zero tilt, and $\theta$ is the tilt angle. Given that the finger-like specimen holder is prone to errors similar to those of other rotating tools, it is necessary to consider the geometric accuracy of the rotation axes. The literature describes four types of errors: axial, face, radial, and tilt error motion[66]. The axial error is not relevant in our system as it is corrected using the x-coordinate of the specimen holder. Face error is specific to machining tools and does not affect the tilt stage. Radial and tilt errors, on the other hand, are addressed together as they result in a circular motion of radius $R$ around the tilt axis. To account for these error motions, a more precise equation is used:

$$a = R * (1 - sin(\theta) + y_A * (1 - cos(\theta) + z_A * sin(\theta))) \tag{2}$$

The protocol for placing point $A$ at the eucentric position is therefore as follows:

- 1. Selection and Settings: Configure the microscope, stage, and image parameters to obtain a clear view of the feature of interest (point $A$) for a tilt angle of 0°. It is important to ensure that the main region of interest (to be studied in 3D) and point $A$ are aligned along the same x-axis.
- 2. Image Acquisition: Acquire a series of images with increasing tilt angles using the microscope. It is essential to find a balance between acquiring a sufficient number of images and the time required for acquisition. To enable effective image matching, it is necessary to select an appropriate tilt angle step size that allows for both speed and the presence of common objects in two consecutive images. A step size of 1-2° is typically adequate. The final tilt angle can be set as the maximum tilt range of the specimen holder, provided that the ROI remains within the field of view throughout the entire tilt range. For image parameters, lower resolution images are sufficient for image matching and offer greater resistance to noise. In our case, for eucentric positioning, we used $512 \times 442$ (ESEM) or $512 \times 512$ (ETEM) pixel images with an exposure time of 1–10 µs.
- 3. Matching Analysis with Feature Matching: Feature matching is a viable option for determining the relative displacement between two images. Well-parameterized feature matching algorithms offer faster computational performance compared to cross-correlation algorithms. The resource-intensive aspect lies in defining the features within the image, which can be delegated to a parallel task. Once the features are defined, matching the labeled images takes only a few milliseconds. Feature matching algorithms demonstrate particular efficacy when the fiducials deposited on the TEM grid are abundant and clearly visible. Moreover, feature matching algorithms exhibit insensitivity to pixel intensity and perspective changes, and can determine the homography between the original image and the target image. In our implementation, we utilized the SIFT features provided by the OpenCV module, with a Flann-based matcher to define the necessary functions for feature matching.

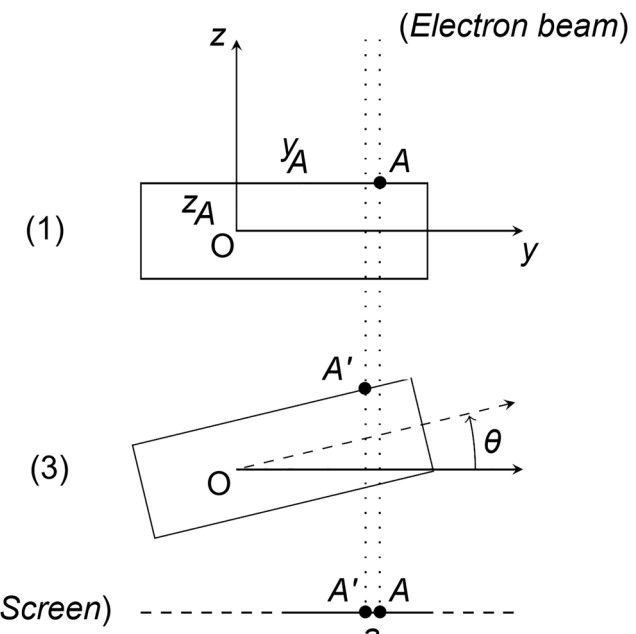

**Fig. 7 | Schematic of the object holder.** The figure presents the position at two tilt angles (0 and $\theta$) and the screen projection.

- 4. Model Fitting: In this computational step, we fit the experimental linear displacement to the theoretical model described earlier (equation (2)). Knowing the displacement a from image matching and the corresponding angles $\theta$, we use the nonlinear least squares function "curve fit" provided by the Python library "SciPy" to apply the fitting procedure and obtain estimates for the parameters $y_A$ and $z_A$.
- 5. Position correction: To achieve precise alignment, we perform a position correction by adjusting the stage relative to the sample holder in the y- and z-directions. Specifically, we move the stage by $-y_A$ and $-z_A$ distances, respectively. Additionally, a stage movement of $y_A$ is executed to ensure the calibration object remains within the field of view. Furthermore, the focus is adjusted from $z_A$ to maintain optimal focus on the sample.
- 6. Validation: To validate the calibration process, tilting experiments are performed at various angles. If the parameter $z_A$ is inappropriately adjusted, the point of interest exhibits lateral displacement when tilted from the 0° angle, aligning with the turnaround position. In addition, if the parameter $y_A$ is not accurately set, point $A$ initially moves in a specific direction, depending on sign of the tilt angle's orientation from the 0° position. Ultimately, if both $z_A$ and $y_A$ are properly configured, the point of interest remains stationary on the screen, confirming the successful calibration.

### Predictive in situ drift correction

To address drift in real time, the algorithm is based on a function named $get\_shift$ ($img_1$, $img_2$), returning the pixel shift between the frame at step $n-1$ and the frame at step $n$. The function gives the relative shift between two images in pixels. At step $n+1$, we determine the shift $d$ as follows:

$$d_n = get\_shift(img_n, img_{n-1}) \tag{3}$$

Subsequently, we compute the correction term $c$ to compensate for the drift:

$$c_n = c_{n-1} - d_n \tag{4}$$

Here, $c_{n-1}$ represents the memory of the previous correction, while $d_n$ is the correction term for the current movement. After applying the

correction, we anticipate the next move by assuming it will follow the previous pattern. This anticipation $a$ is calculated as:

$$a_n = a_{n-1} + c_n \quad (5)$$

Similarly to before, $a_{n-1}$ stores the memory of the previous anticipation, and $c_n$ represents the anticipation subterm for the current step. Finally, we calculate the final relative beam shift $r_n$ to be applied. To ensure consistency, a conversion step is performed to convert lengths from pixels to international units.

$$\begin{cases} r_n = a_n + c_n \\ r_n = a_{n-1} + 2*c_{n-1} - 2*d_{n-1} \end{cases} \quad (6)$$

Supplementary Fig. 2 illustrates the graphical block diagram of the entire process. The input corresponds to the sample position at step $n$, while the output represents the sample position at step $n + 1$. For the initial position at $n = 0$, we assume it to be 0 since we track the object in the center of the image. Additionally, $c_0$ and $a_0$ are both initialized as 0. The Drift Correction (D.C.) point is responsible for executing the correction and anticipation steps. The feedback loop takes into account any potential unknown disturbances caused by the aforementioned mechanisms, incorporating their effects in the subsequent $n + 1$ step.

### Correcting drift in z-direction

During tilting, drift can also occur along the z-direction, leading to image blur by defocus. Moreover, inadequate beam alignments or electron charging effects can induce astigmatism[67]. Correcting this blur requires a similar approach, with the challenge of determining the direction of focus and stigmatism change. Indeed, in STEM mode, the absence of true Fresnel fringes renders blurred images indistinguishable in both directions[68]. It is thus essential to establish a means of quantifying focus and astigmatism, frame by frame, and relate these variations to defocus distance and astigmatism levels. Again, the introduction of a validation step between two successive image acquisitions should be avoided. Therefore, accurate estimation, correction, and anticipation of variations should be accomplished within a single step, based on previous data. Two distinct approaches have been identified in the literature. The first involves computing the variance of a Laplacian of a Gaussian filtered image[69], while the second method relies on estimating the elliptical shape of the Fast Fourier Transform (FFT)[70]. Both methods offer advantages in different scenarios. While live focus and astigmatism correction are still being perfected, the software displays to the user a live FFT of the acquired image, overlaid with an indication of the estimated focus and stigmator drifts. This information allows the user to adjust these parameters manually on the fly.

### Water conservation throughout the experimental process

Ensuring the presence of water throughout the experimental process, from droplet deposition to the completion of acquisition, despite potential fluctuations in pressure and temperature, presents a recurring challenge. Two simple complementary indicators, namely hydration conditions and visual identification, can be used to confirm the presence of liquid water in both microscopes.

In ESEM, the hydration conditions are constantly ensured as long as the sample temperature is stable and the pressure never falls below the saturating vapor pressure - corresponding to 100% relative humidity (RH). Conversely, in ETEM, the initial step of introducing the cryogenic holder under vacuum necessitates a reliable seal of the tab protecting the sample. For samples resistant to moderate freezing, temporary freezing below 0° can be used until pressure conditions compatible with the liquid state of water are reached.

In addition, visual identification is useful to confirm the presence of water, and can take various forms. Initially, the presence of a sufficiently thick layer of water that is not electron-transparent is a proof that the sample is completely hydrated. After water layer thinning, the presence of micro-droplets of water on the TEM grid membrane can be another piece of evidence of the adequate hydration state. Finally, once the acquisition is complete, a series of images can be acquired during or after dehydration to confirm the disappearance of water and visualize associated changes. Although each observation alone does not guarantee the continuous presence of water on the studied particle during each stage of acquisition, the whole set of observations collectively provide reasonable confidence in the hydration state, taking into account the stability of the experimental setup, in particular the temperature. Supplementary Figs. 3, 4 and 5 show examples of the presence of water in the case of the aluminium hydroxide gel studied below.

### Bacterial cultures

*Magnetospirillum magneticum* AMB-1 (ATCC 700264) was cultivated following the protocol described by Komeili and co-workers[48]. The sole iron source provided to AMB-1 cultures corresponded to Fe(III)-citrate at 100 µM. Cells were harvested during late exponential phase. To minimize precipitation of salts from ions contained in the growth medium during liquid STEM analyses, cells were centrifuged (15 min, 4000 rpm), transferred into diluted fresh growth medium (i.e., one volume of medium in two volumes of Milli-Q water), and loaded onto the microscopy grid. For conventional STEM, AMB-1 cells were deposited on carbon-coated copper grids and analyzed with a IT800HL scanning electron microscope (JEOL) operating at 30 kV.

### Reporting summary

Further information on research design is available in the Nature Portfolio Reporting Summary linked to this article.

### Data availability

The data that support the findings of this study are available from the corresponding author upon reasonable request.

### Code availability

The software M-SIS is available on GitHub at https://github.com/louim-lbs/Process_Integration.

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

## Acknowledgements

This work was funded by the French National Research Agency ANR (WATEM Project ANR-20-CE42-0008-02 for the acquisition of the cryo holder, and Project Liquid 3D-STEM ANR-20-CE92-0014-01 for the PhD scholarship and all other developments, and Project BIOMAG ANR-23-CE49-0007 for bacterial cultures and preparation of the biological samples). We acknowledge the Consortium Lyon - St-Étienne de Microscopie (CLYM) for the access to the ESEM ThermoFisher Quattro and ETEM FEI Titan electron microscopes. We thank Niels de Jonge and Diana Peckys for fruitful discussion.

## Author contributions

This work corresponds to the PhD work of L.M.L., done under the supervision of K.M.V. and L.R., thanks to the funds obtained by K.M.V. J.F., E.A. and C.G. designed and built the stage for ESEM experiments according to the needs expressed by L.M.L., L.R. and K.M.V. C.M. gave advice to L.M.B. and did specific developments on TomoJ. M.Ao., L.B. and V.T. gave a precious help to L.M.L. and L.R. for the experiments in ESEM and ETEM. M.Am. cultured the bacteria and supervised the experiments on bacteria. V.N., C.F. and V.T. were also involved into the experiments on bacteria. All co-authors worked on the paper: a first version of the article was written by L.M.L., modified by L.R. and K.M.V., then revised by all the co-authors.

## Competing interests

The authors declare no competing interests.
