## [Transparent Peer Review file · Communications Engineering]

Fast automatic multiscale electron tomography for sensitive materials under environmental conditions

Corresponding Author: Dr Louis-Marie Lebas

Version 0:

Reviewer comments:

Reviewer #1

(Remarks to the Author)

This paper Lubas et al. presents a novel approach to automating 3D tilt series ET (electron tomography) for sensitive materials. The authors describe a method that combines advancements in software (M-SIS) and hardware (sample holder), improving the speed, accuracy, and efficiency of this imaging technology, applicable in both ESEM (Environmental Scanning Electron Microscopy) and ETEM (Environmental Transmission Electron Microscopy). The authors showcase the capabilities of their technique by imaging an aluminum hydroxide hydrogel in both ESEM and ETEM, showing how M-SIS controls electron dose to limit sample damage. This paper presents a promising technique for imaging electron-beam-sensitive samples in liquid environments, and the methodology and results could have significant impacts on fields like material science and biology. The paper is well written, and the quality of the figures is high. I have only one comment:

- This manuscript focuses on sensitive materials under environmental conditions, concentrating on experiments with aluminum hydroxide hydrogel, but doesn't provide examples of actual biological samples. If experiments involving biological samples (such as cells or viruses) were included (even briefly), it would greatly enhance the reliability of this technology's applicability in biological contexts.

Reviewer #2

(Remarks to the Author)

The manuscript describes procedures and software control that enables efficient image acquisition in an environmental TEM. Results obtained using a beam-sensitive gel sample system are described in order to illustrate operation of the system. Useful resolution is obtained in STEM mode within minutes and with an acceptable electron dose.

The procedures are described in some detail and will be of interest to specialists concerned with the practical details of tomography, particularly with beam-sensitive samples. The meaning is generally clear but from time to time the English is not easy to understand.

I can recommend publication subject to attention to the following comments.

Line 42: "STEM provides high spatial resolution and contrast for biological materials, with a lower electron dose rate compared to conventional TEM due to controlled dwell time [8]." Is the dose rate lower in STEM? Does "due to controlled dwell time" refer to the electron dose rate or to resolution and contrast?

Line 44: "However, the cryogenic method limits the study of liquid water samples under dynamic or environmental conditions," Does this mean that the sample should be liquid, not frozen?

Fig 1, Fig.2, Fig.3: lettering is very small and hard to read. There is too much information compressed into each of these Figures. I recommend separating them.

Line 153: "Results in Supplementary Information Figure 2 demonstrate our solution effectively corrects drift, keeping the ROI centered." If the main purpose of the paper is to demonstrate success of the drift correction, this Figure should be included in the main text.

Version 2:

Reviewer comments:

Reviewer #1

(Remarks to the Author)

The revised manuscript has addressed my previous comment and has been improved. The newly added data further strengthen the value of the authors' research and technique, and will be of greater benefit to readers. The new data and their accompanying description have been appropriately integrated into the manuscript. I have no further comments.

Reviewer #2

(Remarks to the Author)

The authors have addressed my concerns, resulting (I think) in a better publication.

The TEM/STEM ambiguity arose because of the difference between instantaneous dose rate and time-averaged dose rate.

The authors' answers below are in blue.

Reviewers' comments:

Reviewer #1 (Remarks to the Author):

This paper Lubas et al. presents a novel approach to automating 3D tilt series ET (electron tomography) for sensitive materials. The authors describe a method that combines advancements in software (M-SIS) and hardware (sample holder), improving the speed, accuracy, and efficiency of this imaging technology, applicable in both ESEM (Environmental Scanning Electron Microscopy) and ETEM (Environmental Transmission Electron Microscopy). The authors showcase the capabilities of their technique by imaging an aluminum hydroxide hydrogel in both ESEM and ETEM, showing how M-SIS controls electron dose to limit sample damage. This paper presents a promising technique for imaging electron-beam-sensitive samples in liquid environments, and the methodology and results could have significant impacts on fields like material science and biology. The paper is well written, and the quality of the figures is high. I have only one comment:

- This manuscript focuses on sensitive materials under environmental conditions, concentrating on experiments with aluminum hydroxide hydrogel, but doesn't provide examples of actual biological samples. If experiments involving biological samples (such as cells or viruses) were included (even briefly), it would greatly enhance the reliability of this technology's applicability in biological contexts.

We appreciate the reviewer's feedback. We appreciate the reviewer's comment about including biological samples. This manuscript focuses on the methodology and its application to aluminum hydroxide hydrogel as a representative system. We understand how important it is to show how the technique can be used with real biological samples. Because of this, we decided to do new experiments. We added a new section to the paper called "Analysis of Hydrated Bacteria." This section shows that electron tomography can be performed on a magnetotactic bacterium in its native state, i.e. with neither dehydration, freezing, fixation or staining. We give experimental details for its successful analysis. In the "Discussions" section, we show a cross-section through the bacterium's volume and analyze the results in light of conventionally obtained images (for instance, the contrast of the PHA inclusions is not inverted as it is in cryoTEM). With our technique, we can see features as small as 15 nanometers in such relatively thick, beam-sensitive, low-contrast biological object.

Reviewer #2 (Remarks to the Author):

The manuscript describes procedures and software control that enables efficient image acquisition in an environmental TEM. Results obtained using a beam-sensitive gel sample system are described in order to illustrate operation of the system. Useful resolution is obtained in STEM mode within minutes and with an acceptable electron dose.

The procedures are described in some detail and will be of interest to specialists concerned with the practical details of tomography, particularly with beam-sensitive samples. The meaning is generally clear but from time to time the English is not easy to understand.

We thank the reviewer for his assessment of our work and his recognition of the methodology's potential impact in advancing imaging techniques for sensitive materials. We hope the doubts of the reviewer

regarding the feasibility of the beam sensitive sample are lifted once we proved that the analysis of hydrated biological sample is possible.

We carefully checked the English in the text and made a few changes to increase the understanding.

I can recommend publication subject to attention to the following comments.

Line 42: "STEM provides high spatial resolution and contrast for biological materials, with a lower electron dose rate compared to conventional TEM due to controlled dwell time [8]. "Is the dose rate lower in STEM? Does "due to controlled dwell time" refer to the electron dose rate or to resolution and contrast?"

- Yes, the dose rate in STEM can be lower compared to conventional TEM when achieving the same final cumulative dose. This advantage allows STEM to mitigate reversible electron-water interactions more effectively, resulting in less cumulative damage to the sample. This is particularly beneficial for studying radiation-sensitive biological materials.
- The sentence "due to controlled dwell time" directly refers to the electron dose rate, as the dwell time determines how long the electron beam interacts with each point of the sample, affecting the dose delivered. While resolution and contrast are linked to the dwell time, they are more of a consequence of the optimized dose rate and signal acquisition.

For clarity, we suggest a revision: "*STEM provides high spatial resolution and contrast for biological materials. It achieves a lower electron dose rate compared to conventional TEM, for the same final cumulative dose, by enabling precise control over the dwell time at each scanned point, which minimizes sample damage and optimizes imaging conditions [8].*"

Line 44: "However, the cryogenic method limits the study of liquid water samples under dynamic or environmental conditions," Does this mean that the sample should be liquid, not frozen?

- The sample in cryo-EM is frozen, which poses a limitation when studying samples suspended in liquid or samples that are retaining liquid such as water. Indeed for some applications in materials science, it is interesting to follow the sample evolution during dehydration. We suggest changing "limits" to "prevents" to avoid introducing wrong interpretation.

Fig 1, Fig.2, Fig.3: lettering is very small and hard to read. There is too much information compressed into each of these Figures. I recommend separating them.

- We have increased the lettering size to improve readability.
- Due to the limitation on the number of figures and given the recommendation to include SI Fig. 2 in the main text (see below), we propose separating only Fig. 1 into two figures.

Line 153: "Results in Supplementary Information Figure 2 demonstrate our solution effectively corrects drift, keeping the ROI centered." If the main purpose of the paper is to demonstrate success of the drift correction, this Figure should be included in the main text.

- The focus of the paper goes beyond drift correction, but since it is a key aspect, we have included SI Fig. 2 in the main text as new Fig. 4.